LMU-ASC 21/25

# Yang-Mills Theory and the $\mathcal{N} = 2$ Spinning Path Integral

**Carlo Alberto Cremonini,**[a] **and Ivo Sachs**[a]

[a] *Arnold-Sommerfeld-Center for Theoretical Physics, Ludwig-Maximilians-Universität, München, Theresienstr. 37, D-80333 München, Germany*

*E-mail:* carlo.alberto.cremonini@gmail.com, ivo.sachs@physik.lmu.de

ABSTRACT: We embed the perturbative Fock state of the Yang-Mills BV-multiplet in the vertex operator algebra of the path-integral for the $\mathcal{N} = 2$ supersymmetric world line and evaluate the pull-back of the latter to an integral form on supermoduli space. Choosing a suitable Poincaré dual on the latter, we show that this integral form describes an extension of Yang-Mills theory. Upon projection back to the Fock space, we recover the Yang-Mills action from the world line. This furthermore gives an a priori justification for the construction of Yang-Mills equations of motion as emerging from deformations of the BRST differential.

## 1   Introduction

Background independence is an important question in string theory and in string field theory (SFT) alike (e.g. [1–4]). Being constructed perturbatively around a given background (e.g. [5–8]), the explicit form of the action will depend on the latter. Indeed, in the standard construction, the kinetic terms, as well as the vertices, are described in terms of a correlation function in a world-sheet conformal field theory (CFT) that itself depends on the background [5]. An alternative approach is to consider a family of world sheet theories that are parametrized by background fields[1] [9–11]. As the background varies, the BRST operator of the corresponding world-sheet varies as well, and one may alternatively consider the deformation problem, by background fields of a given BRST operator $Q$ as an element of a given graded differential algebra [9, 12, 13], acting on the

---

[1]Although it is not clear how to deform the world sheet theory by a generic massive string field.

appropriate vector space (e.g. [14, 15]). However, this is difficult to carry through explicitly beyond infinitesimal perturbations.

On the other hand, for the spinning world line, as an infinite tension limit of the super string, this approach was successful [16] and it is possible to show that that the full, non-linear Yang-Mills equations of motion are encoded in the nilpotency of $Q$ acting on a suitable restriction $V_0$ of the perturbative space of states[2] $V$ of the spinning world line with $\mathcal{N} = 2$ supersymmetry. Furthermore, this result extends to all massless fields in the NS-sector of closed super string field theory [17]. One then faces the question of how the perturbative construction of SFT relates to the deformation problem. The first construction is usually based on a (non)-polynomial SFT-action whose perturbative vertices, in the operator formalism, have an interpretation as $n$-ary maps $l_n$ on $V^{\otimes n}$ with the structure of an $L_\infty$ algebra. One attempt to relate this construction to the deformation problem is to identify deformations of $Q$ with $\{l_n\}$ via the operator-state map [13, 16]. However, as explained in [13, 18], this identification fails for the spinning world line due to the failure of the operator-state map being an isomorphism[3] on $V_0$. For the same reason, we do not know how to construct an action $S[Q]$ [19] from which the nilpotency of $Q$ derives as an equation of motion.

This provides a motivation for the present paper, where we construct the vertices of the space-time action for Yang-Mills theory as an integral form of world-line $n$-point functions on the supermoduli space, $\mathcal{M}$ of the $\mathcal{N} = 2$ spinning particle, thus mimicking the construction of super string field theory (e.g. [6, 20, 21]). To this end, we first embed the perturbative BRST spectrum of Yang-Mills theory into the algebra of vertex operators of the world line as a state-operator map. As mentioned above, this map is not unique. We determine it by demanding that the embedding defines a quasi-isomorphism to a suitable sub-complex of $\mathcal{N} = 2$ vertex operator algebra. For the quadratic space-time action derived from the world-line path integral, this amounts to "integrating in" auxiliary fields whose elimination implements the quasi-isomorphism.[4]

In the next step, we derive the cubic interaction vertex as a pull-back of the (off-shell) 3-point function to $\mathcal{M}$. This requires a choice of a Poincaré dual, $Y$ on $\mathcal{M}$ (see, e.g., [22] where this object is introduced as a cochain map between different complexes on a supermanifold), since the latter is of odd dimension two rather than one, which is the expected dimension for Yang-Mills theory [20, 23]. We will show that different choices of $Y$ correspond to field redefinitions in the space-time action. Furthermore, we show that there is a choice of $Y$ for which the 3-point function reproduces either the perturbative cubic interaction term, or the linear deformation of the BRST operator $Q$ in [16]. Which object is derived depends on how picture changing is implemented on the 3rd vertex (see [24] and [22] for a description of the picture changing operation). This is in agreement with

---

[2] In SFT one is not guaranteed that $V$ itself does not depend on the background.

[3] In string theory, the operator-state map is (formally) an isomorphism. However, the reduction to $V_0$ is not compatible with it as a result of the inconsistency of level truncation. This renders the deformation problem of $Q$ difficult to handle.

[4] This is just the opposition of the common usage of homotopy transfer, where one usually integrates out fields to obtain an effective action.

the realization found in [23] for the $\mathcal{N} = 1$ world line. In addition to the standard cubic interaction, the pull-back of the path integral features additional interactions with higher form potentials that are inherently present in the vertex operator algebra, in agreement with the $\mathcal{N} = 1$ world line [23]. However, in $\mathcal{N} = 2$ we have the possibility to project the external states onto the Fock space of the perturbative Yang-Mills multiplet. In doing so, we recover the deformed BRST operator of [16] in a background. This establishes the equivalence of the deformation problem of $Q$ and the polynomial action derived from the world-line, which, however, is not realized as a homotopy associative algebra on a Fock space $V$ but rather a cyclic complex as in [25].

The geometric decomposition of $\mathcal{M}$ for four punctures suggests an additional quartic vertex, and indeed gauge invariance requires this too. In the world line theory, it can be realized by adding infinitesimal stubs to the cubic vertex together with a "small" bosonic modulus for the quartic vertex. Integration over the odd directions of $\mathcal{M}_4$ results in a total derivative for the bosonic modulus, leaving us with a boundary term that results in the expected quartic contact vertex on Yang-Mills theory. This gives a geometric interpretation of the presence of the quartic contact term arising in the construction of super string field theory [7]. Higher order contact terms do, however, not arise on the world line. This property can be derived directly from the geometry of $\mathcal{M}$.

## 2   $\mathcal{N} = 2$ Spinning Particle

We will consider the BRST quantization of the $\mathcal{N} = 2$ spinning particle in the Hamiltonian formulation. We recast the two worldline spinors into complex linear combinations of $\psi^\mu = \frac{1}{\sqrt{2}}(\psi_1^\mu + i\psi_2^\mu)$, $\bar{\psi}^\mu = \frac{1}{\sqrt{2}}(\psi_1^\mu - i\psi_2^\mu)$. After gauge fixing the Einbein $e = 1$ and the world line gravitino $\chi = 0$, the BRST-invariant world-line action is then given by

$$I = \int p\dot{x} - \frac{1}{2}p^2 + i\bar{\psi}\dot{\psi} + ib\dot{c} - i\bar{\beta}\dot{\gamma} - i\beta\dot{\bar{\gamma}} \tag{2.1}$$

which gives the fermionic 2-point function

$$\langle \bar{\psi}^\mu(t)\psi^\nu(t')\rangle = \frac{1}{2}\text{sgn}(t - t')g^{\mu\nu}, \quad \langle p_\mu(t)x^\nu(t')\rangle = \frac{-i}{2}\text{sgn}(t - t')\delta^\nu_\mu,$$

$$\langle b(t)c(t')\rangle = \langle \beta(t)\bar{\gamma}(t')\rangle = \langle \bar{\beta}(t)\gamma(t')\rangle = \frac{1}{2}\text{sgn}(t - t'). \tag{2.2}$$

In the operator formulation, this gives rise to the relations

$$[\beta, \bar{\gamma}] = [\bar{\beta}, \gamma] = 1, \quad \{\psi^\mu, \bar{\psi}^\nu\} = g^{\mu\nu}, \tag{2.3}$$

and the world-line BRST operator is given by

$$Q = cH + \bar{\gamma}q + \gamma\bar{q} + \gamma\bar{\gamma}b, \tag{2.4}$$

where $q = \psi^\mu p_\mu$, $\bar{q} = \bar{\psi}^\mu p_\mu$ and $H$ generate the (local) SUSY transformations and reparametrizations, respectively.

This model then describes the BRST-spectrum of Yang-Mills theory as a state $|\Phi\rangle$ in a suitable world-line Fock module generated by $\psi, \beta, \gamma$ and $c$ with coefficients in smooth (wave) functions. That is

$$|\Phi\rangle = \left( A_\mu(x)\psi^\mu + \beta C(x) + c\beta i B(x) \right) \delta(\bar{\gamma}) 1_\psi 1_c 1_\gamma \in V^{(0,-1)} 1_\psi \,, \tag{2.5}$$

where the *picture* $(0, -1)$ stands for the super ghost ground state $1_\psi \delta(\bar{\gamma})$. Here, the conjugate momenta act from the left as derivatives, i.e.

$$p_\mu = -i\partial_{x^\mu} \,, \qquad \bar{\psi}_a = \partial_{\psi^a} \,, \qquad b = \partial_c, \qquad \bar{\gamma} = -\partial_\beta, \qquad \bar{\beta} = \partial_\gamma \,. \tag{2.6}$$

Furthermore,

$$S = \int \langle \Phi | Q | \Phi \rangle d^D x \tag{2.7}$$

reproduces the familiar BRST-extended space-time action for Maxwell theory.

More generally, [13, 16] consider the deformation problem of $Q$ on the invariant subspace $V^{(0,-1)} 1_\psi$ defined as the eigenspace of the R-charge

$$J = \psi^\mu \bar{\psi}_\mu + \gamma \bar{\beta} - \beta \bar{\gamma} + 1 = N_\psi + N_\gamma + N_\beta + 1 \,, \tag{2.8}$$

with charge 2. It was then observed in [16] that on $V^{(0,-1)} 1_\psi$ the deformed BRST-operator $Q(A)$ with

$$q = \psi^\mu \Pi_\mu \,, \qquad \bar{q} = \bar{\psi}^\mu \Pi_\mu \,, \qquad H = -\Pi^2 - 2\kappa \psi^\mu \bar{\psi}^\nu [\Pi_\mu, \Pi_\nu] \,, \qquad \kappa \in \mathbb{R} \,, \tag{2.9}$$

with $\Pi_\mu = p_\mu + A_\mu$ is nilpotent provided $\kappa = 1$ and $A$ satisfies the non-abelian equations of motion for Yang-Mills theory. We end this review with two comments. 1) Notice that the last term in the above equation maps to 0 in $V^{(0,-1)} 1_\psi$ by the operator-state map. It can thus not be reproduced by a multi-linear map on $V^{(0,-1)} 1_\psi$. Therefore, the non-linear Yang-Mills equation cannot be written as a MC-equation on $V^{(0,-1)} 1_\psi$. 2) Here, choosing $\kappa = 1$ is motivated by the desire to recover the Yang-Mills equation of motion. However, in a SFT description, this should follow from a world-line calculation. We will address these points in the following sections.

## 3 State-Operator Map

The first step in our construction is to identify a suitable subcomplex of the full $\mathcal{N} = 2$ world-line vertex operator algebra $\left( V^{(0,-1)}, Q \right)$ which contains the interaction term in (2.9), but encodes the same physical states as the representation (2.5) being quasi-isomorphic to it. We thus seek a cochain map $\iota$

$$\left( V^{(0,-1)} \, 1_\psi, Q \right) \xrightarrow{\ \iota\ } \left( V^{(0,-1)}, Q \right) \tag{3.1}$$

such that $\iota \left( V^{(0,-1)} \right)$ is quasi-isomorphic to $V^{(0,-1)} 1_\psi$.

**Remark:** On can similarly remove the $\mathbf{1}_x$ ground state from $V$ which, upon acting with $Q$

will result in vertex operators that are polynomial on $p_\mu$. In fact, this will be the approach that we will follow when considering the interaction terms in section 5. For the quadratic action that encodes the linear cohomology, keeping $\mathbf{1}_x$ or not, is equivalent up to total derivatives. In this section, we keep $\mathbf{1}_x$ to simplify the construction of the embedding.

Simply identifying the operators acting on the vacuum with themselves, without the insertion of other terms projected out by $1_\psi$, would not lead to a map commuting with the differential. For example, for the piece of the vertex given by $c\beta i B(x)\delta(\bar\gamma)$, we have (we consider here only the term linear in $c$)

$$[Q(c\beta i B(x)\delta(\bar\gamma)1_\psi)]_c = [-\psi^\mu c\, p_\mu i B(x)\delta(\bar\gamma)]\, 1_\psi \xmapsto{\iota} -\psi^\mu c\, p_\mu i B(x)\delta(\bar\gamma)\ . \tag{3.2}$$

On the other hand, if we first apply $\iota$, we have

$$c\beta i B(x)\delta(\bar\gamma)1_\psi \xmapsto{\iota} c\beta i B(x)\delta(\bar\gamma) \xmapsto{Q|_c} -\psi^\mu c p_\mu i B(x)\delta(\bar\gamma) + \bar\psi^\mu c\gamma\beta p_\mu i B(x)\delta(\bar\gamma)\ . \tag{3.3}$$

This suggests that, when removing the ground state $1_\psi$, we need to enlarge the given minimal set of fields. We thus extend the initial set of fields (we do not display anti-fields here)

$$\left(A_\mu\psi^\mu + \beta C + c\beta i B\right)\delta(\bar\gamma) \tag{3.4}$$

by including all the other fields generated by the action of $Q$. This gives an extended multiplet of the form

$$\begin{aligned} &A_\mu\psi^\mu + \beta C + c\beta i B \\ &- \beta\gamma A_\mu\bar\psi^\mu + \gamma i B^* + \gamma i B^*_{\mu\nu}(\bar\psi^\mu\psi^\nu + \psi^\mu\bar\psi^\nu) - \beta\gamma^2 Z^*_{\mu\nu}\bar\psi^\mu\bar\psi^\nu + \cdots\ , \end{aligned} \tag{3.5}$$

where the $\cdots$ contains anti-fields and terms generated upon action with $Q$ on the other terms as well. Notice that we have identified the space-time field multiplying $\beta\gamma\bar\psi^\mu$ with the gauge field $A_\mu$. This is suggested by the action of $Q$ on $\beta C$, implying that the space-time fields multiplying $\psi^\mu$ and $\beta\gamma\bar\psi^\mu$ transform identically[5]. The same argument will be used for the other identifications later. Thus we have

$$\iota\left(A_\mu\psi^\mu\delta(\bar\gamma)1_\psi\right) = A_\mu\psi^\mu\delta(\bar\gamma) - \beta\gamma A_\mu\bar\psi^\mu\delta(\bar\gamma)\ . \tag{3.6}$$

However, this embedding does still not commute with $Q$. Indeed, acting with $Q$ on (3.6) and focusing on terms proportional to $\gamma$ we obtain

$$\left(p_\mu A_\nu\bar\psi^\mu\psi^\nu + p_\mu A_\nu\psi^\nu\bar\psi^\mu\right)\gamma\delta(\bar\gamma) = \left[-i(dA)_{\mu\nu}(\psi^\mu\bar\psi^\nu + \bar\psi^\mu\psi^\nu) - i\delta A\right]\gamma\delta(\bar\gamma)\ . \tag{3.7}$$

On the other hand, one has (again, in the $\gamma$ sector)

$$[Q(A_\mu\psi^\mu\delta(\bar\gamma)1_\psi)]_\gamma = -i\delta A\gamma\delta(\bar\gamma)1_\psi \xmapsto{\iota} -i\delta A\gamma\delta(\bar\gamma)\ . \tag{3.8}$$

---

[5]Making this identification fixes the 1-form ghost present in this BV-multiplet.

Here, we have chosen the map $\iota$ to be just the identification for fields proportional to $\gamma$. We can then compensate for the extra $dA$ by further modifying $\iota$ as

$$\iota\left(A_\mu\psi^\mu\delta(\bar\gamma)1_\psi\right) = A_\mu\psi^\mu\delta(\bar\gamma) - \beta\gamma A_\mu\bar\psi^\mu\delta(\bar\gamma) - ic\beta(dA)_{\mu\nu}(\psi^\mu\bar\psi^\nu + \bar\psi^\mu\psi^\nu)\delta(\bar\gamma)\ , \qquad (3.9)$$

such that now the action of $Q$ in the $\gamma$ sector gives $\delta A$ only, as in (3.8).

The introduction of the last term in (3.9) fixes the cochain map property at linear order in $\gamma$, but further terms are needed. For example, at order $c$ we have

$$\left[Q\left(A_\mu\psi^\mu\delta\left(\bar\gamma\right)1_\psi\right)\right]_c = c\Box A_\mu\psi^\mu\delta\left(\bar\gamma\right)1_\psi \overset{\iota}{\mapsto} c\Box A_\mu\psi^\mu\delta\left(\bar\gamma\right)\ , \qquad (3.10)$$

while on the other hand, one has

$$\left[Q\left(A_\mu\psi^\mu\delta(\bar\gamma)-\beta\gamma A_\mu\bar\psi^\mu\delta(\bar\gamma) + c\beta(dA)_{\mu\nu}(\psi^\mu\bar\psi^\nu + \bar\psi^\mu\psi^\nu)\delta(\bar\gamma)\right)\right]_c$$
$$= c\Box A_\mu\psi^\mu\delta(\bar\gamma) - icq(dA)_{\mu\nu}(\psi^\mu\bar\psi^\nu + \bar\psi^\mu\psi^\nu)\delta(\bar\gamma)\ . \qquad (3.11)$$

This shows that the map $\iota$ is not a cochain map yet. It needs to be modified with an extra term. The latter can be expressed as a non-local contribution

$$\iota\left(A_\mu\psi^\mu\delta(\bar\gamma)1_\psi\right) = A_\mu\psi^\mu\delta(\bar\gamma) - \beta\gamma A_\mu\bar\psi^\mu\delta(\bar\gamma) - c\beta(dA)_{\mu\nu}(\psi^\mu\bar\psi^\nu + \bar\psi^\mu\psi^\nu)\delta(\bar\gamma)$$
$$+ iq(\frac{d}{\Box}A)_{\mu\nu}(\psi^\mu\bar\psi^\nu + \bar\psi^\mu\psi^\nu)\delta(\bar\gamma)\ . \qquad (3.12)$$

Summing up, the previous discussion suggests that we can add to the vertex terms of the following form

$$c\beta iB_{\mu\nu}\bar\psi^\mu\psi^\nu + c\beta iB_{\mu\nu}\psi^\mu\bar\psi^\nu - \frac{1}{2}c\beta^2\gamma iB_{\mu\nu}\bar\psi^\mu\bar\psi^\nu$$
$$+ cA^*_{\mu\nu\rho}\psi^\mu\bar\psi^\nu\psi^\rho + cA^*_{\mu\nu\rho}\psi^\mu\psi^\nu\bar\psi^\rho - c\beta\gamma A^*_{\mu\nu\rho}\bar\psi^\mu\bar\psi^\nu\psi^\rho - c\beta\gamma A^*_{\mu\nu\rho}\bar\psi^\mu\psi^\nu\bar\psi^\rho$$
$$- c\beta\gamma A^*_{\mu\nu\rho}\psi^\mu\bar\psi^\nu\bar\psi^\rho + \frac{1}{2}c\beta^2\gamma^2 A^*_{\mu\nu\rho}\bar\psi^\mu\bar\psi^\nu\bar\psi^\rho + \qquad\qquad (3.13)$$
$$A_{\mu\nu\rho}\psi^\mu\bar\psi^\nu\psi^\rho + A_{\mu\nu\rho}\psi^\mu\psi^\nu\bar\psi^\rho - \beta\gamma A_{\mu\nu\rho}\bar\psi^\mu\bar\psi^\nu\psi^\rho - \beta\gamma A_{\mu\nu\rho}\bar\psi^\mu\psi^\nu\bar\psi^\rho - \beta\gamma A_{\mu\nu\rho}\psi^\mu\bar\psi^\nu\bar\psi^\rho$$
$$+ \frac{1}{2}\beta^2\gamma^2 A_{\mu\nu\rho}\bar\psi^\mu\bar\psi^\nu\bar\psi^\rho + \cdots\ . \qquad\qquad (3.14)$$

We may assume $B_{\mu\nu}$ to be antisymmetric and that $A_{\mu\nu\rho}$ is antisymmetric in the last two indices because of the structure of the last term of (3.12). Eqn. (3.9) would allow us to further restrict $B_{\mu\nu}$ to be d-exact, but for now, we will keep $B_{\mu\nu}$ unconstrained in order to have a free variation of the action below.

The full consistent multiplet, compatible with the cochain map $\iota$ in (3.1), is constructed by iterating the previous arguments, thus leading to higher fields which we will not list here for brevity. The linearized equations of motion encoded in the $Q$-closure of the vertex without ground state $1_\psi$ imply

$$-i\partial^\alpha A_{\alpha\mu\nu} - i(\mathrm{d}A)_{\mu\nu} - iB_{\mu\nu} = 0\,, \qquad (3.15a)$$
$$\Box A_{\mu\nu\rho} - i\partial_\mu iB_{\nu\rho} - i\partial^\lambda iB_{\lambda\mu\nu\rho} = 0\,, \qquad (3.15b)$$
$$-i\partial_\mu A_{\nu\rho\sigma} - iB_{\mu\nu\rho\sigma} = 0\,, \qquad (3.15c)$$

for the auxiliary fields. This implies, in turn

$$\tilde{\delta}\tilde{\mathrm{d}}A^{(3)} = \tilde{d}\mathrm{d}A^{[1]}, \tag{3.16a}$$

$$\tilde{d}dA^{[1]} = 0, \tag{3.16b}$$

where $\tilde{\mathrm{d}}$ and $\tilde{\delta}$ are used to denote the de Rham operator and its adjoint, but blind to the last two indices of $A^{(3)}$. We use the notation with round brackets $A^{(n)}$ for tensors that are not necessarily totally antisymmetric in all indices, and the notation with square brackets $A^{[n]}$ for totally antisymmetric tensors.

Note that it is consistent to impose the *conventional constraint* $B_{\mu\nu} = 0$, in which case the equations above are equivalent to

$$\tilde{d}\tilde{\delta}A^{(3)} = 0 \quad \text{and} \quad \tilde{\delta}A^{(3)} + \mathrm{d}A^{[1]} = 0. \tag{3.17}$$

It is then consistent to assume $A^{(3)}$ to be totally antisymmetric, i.e. $A^{(3)} = A^{[3]}$ (see appendix A). In that case, we recover the equation of motion of duality invariant Maxwell theory previously derived from the pure $\mathcal{N} = 1$ world line [23]. This is consistent with the observation that in the $Q_{\mathcal{N}=1}$ cohomology, $B_{\mu\nu}$ is absent because this equation is obtained by acting on the picture-zero vertex, which does not contain the field $B^{[2]}$. When deriving the e.o.m. for the $\mathcal{N} = 1$ action as in [23], $B_{\mu\nu}$ is absent because it enters only linearly there, in the form of a Lagrange multiplier.

In sum, we see that the multiplets and gauge symmetries with or without setting $B_{\mu\nu} = 0$ are quite different. In the former case, the multiplet is built out of mixed symmetry tensors, which is substantially bigger and accordingly has a bigger gauge redundancy as well.

## 4 Quadratic Action

Let us now work out the quadratic action from the world-line path integral by inserting two vertices in ghost number 0 on the world line. We have two kinds of vertices, in ghost number 0, one in the $\delta(\gamma)$ (which we will denote $\mathcal{V}^{(-1,0)}$ to indicate one $\delta$ in the $\gamma$ direction and no $\delta$ in the $\bar{\gamma}$ direction) and the other in the $\delta(\bar{\gamma})$ (which we will denote $\mathcal{V}^{(0,-1)}$) representations. The former has $R$-charge $-2$, the latter has $R$-charge $+2$. In the paragraphs above we have constructed the (first pieces of the) vertex $\mathcal{V}^{(0,-1)}$ describing the same degrees of freedom of the $\mathcal{N} = 1$ theory; in order not to introduce new degrees of freedom, the vertex $\mathcal{V}^{(-1,0)}$ is constructed with *the same field content*, that is

$$\mathcal{V}^{(-1,0)} := \mathcal{V}^{(0,-1)} \left( \psi \leftrightarrow \bar{\psi}, \gamma \leftrightarrow \bar{\gamma}, \beta \leftrightarrow \bar{\beta} \right). \tag{4.1}$$

The quadratic part of the action thus reads

$$S^{(2)} = \int d^4x\mathrm{d}c\mathrm{d}\bar{\gamma}\mathrm{d}\gamma \langle \mathcal{V}^{(-1,0)}(-\infty) Q \mathcal{V}^{(0,-1)}(\infty) \rangle \tag{4.2}$$

where $\int d^4x\mathrm{d}c\mathrm{d}\bar{\gamma}\mathrm{d}\gamma$ is over the zero modes,

$$\mathcal{V}^{(-1,0)}(s) = e^{isH}\mathcal{V}^{(-1,0)}(x, \psi, \bar{\psi}, \bar{\gamma}, \bar{\beta}, c)\delta(\gamma)\, e^{-isH}$$

$$\mathcal{V}^{(0,-1)}(s) = e^{isH}\mathcal{V}^{(0,-1)}(x, \psi, \bar{\psi}, \gamma, \beta, c)\delta(\bar{\gamma})e^{-isH}, \tag{4.3}$$

and the angular bracket denotes the expectation value in the world-line quantum field theory evaluated with the action given in (2.1). See also [23] for more details. A lengthy, but otherwise straightforward calculation gives

$$S^{(2)} = \int \left( -\frac{1}{2}(\mathrm{d}A^{[1]}, \mathrm{d}A^{[1]}) + \frac{3}{2}(\tilde{\delta}A^{(3)}, \tilde{\delta}A^{(3)}) + (\delta A^{[1]} + B^{[0]}, \delta A^{[1]} + B^{[0]}) \right. \tag{4.4}$$
$$\left. + \frac{3}{2}(\mathrm{d}A^{[1]} + \tilde{\delta}A^{(3)} + B^{[2]}, \mathrm{d}A^{[1]} + \tilde{\delta}A^{(3)} + B^{[2]}) + \frac{3}{2}(\tilde{\mathrm{d}}A^{(3)} + B^{(4)}, \tilde{\mathrm{d}}A^{(3)} + B^{(4)}) \right) + \cdots$$

where $\tilde{\delta}A^{[3]} = \partial^\alpha A_{\alpha\mu\nu}$ and $(\cdot, \cdot)$ denotes the full contraction of all $\psi$ and $\bar{\psi}$. Variation w.r.t. $B^{[2]}$ and $A^{(3)}$ reproduces the equations (3.15). If we subsequently set $B^{[2]} = 0$, this implements the quasi-isomorphic embedding with the extra field $A^{[3]}$ constructed in the previous section. Furthermore, inserting the projection $\mathbf{1}_\psi$ and $\mathbf{1}_{\bar{\psi}}$ at $s = \pm\infty$ respectively projects out all higher form fields, thus reproducing (2.7) (times $\frac{1}{2}$).

## 5 Interactions

Let us now turn to the cubic interaction vertex in the space-time action, which should come from the pull-back of the world-line 3-point function to the supermoduli space $\mathcal{M}_3$. The introduction of the extra puncture on the world-line then naturally requires a $(0|-2)$-integral form (that is, with ghost number zero and total picture number -2) on $\mathcal{M}_3$ which is tantamount to a double picture changing, thus producing a 2-derivative interaction[6]. This is expected for a theory of gravity, familiar from the $\mathcal{N} = (1, 1)$ world sheet of the closed string, but not for Yang-Mills theory. We therefore introduce a $(0|1)$-dimensional subspace that is identified via its Poincaré dual $Y$, thus "trivialising" one of the two odd moduli we integrate over.[7] If we denote by $i : \mathcal{M}^{(0|1)} \hookrightarrow \mathcal{M}^{(0|2)}$ the embedding of the $(0|1)$-dimensional subspace in the moduli space $\mathcal{M}^{(0|2)}$, we can formally write

$$\int_{\mathcal{M}^{(0|2)}} \omega \cdot Y = \int_{\mathcal{M}^{(0|1)}} i^*\omega , \tag{5.1}$$

where $i^*\omega$ is an integral form on the subspace. If we denote the coordinates corresponding to the two odd moduli by $\eta$ and $\bar{\eta}$, and we want to trivialise, for example, the modulus given by the $\eta$ coordinate, we can choose the representative $Y = \eta\delta(d\eta)$. Thus, the cubic term is of the form

$$S^{(3)} = \int_{\mathcal{M}^{(0|2)}} [\mathrm{d}\bar{\eta}\mathrm{d}\eta | \mathrm{d}(\mathrm{d}\bar{\eta})\mathrm{d}(\mathrm{d}\eta)] \, Y \, \langle \mathcal{V}^{(-1,0)}(-\infty)e^{iF_0} \, \mathcal{V}(0) \, e^{-iF_0}\mathcal{V}^{(0,-1)}(+\infty)\rangle . \tag{5.2}$$

The vertex operators inserted in $\pm\infty$ have the structure (picture number, $R$-charge, etc.) as those described in the previous section. Notice that the map $Y$ has picture number $-1$, which implies that it must be paired with the rest of the integrand with total picture (on the supermoduli space) $-1$ as well. We further require $\mathcal{V}(0)$ to have zero $R$-charge if we

---

[6]Indeed, integration of a odd direction in $\mathcal{M}$ comes with an adjoint action of the supercharge on the vertex operator, see e.g. [23] for more details.

[7]A related construction was proposed in [26, 27] by embedding the $\mathcal{N} = 1$ string in a twisted $\mathcal{N} = 2$ theory.

want to interpret $S^{(3)}$ as a deformation of $Q$. The operator inserted at 0 is thus constrained to have picture number $-1$, ghost number one and $R$-charge zero. In particular, it has a different structure (in terms of $R$-charge) with respect to the vertices appearing in the kinetic term. The adjoint action by $F_0$ implements the pullback to the supermoduli space and will be described in the next section. Before doing so, let us, however, first comment on some properties of the Poincaré dual $Y$.

## 5.1 Dependence on the Poincaré Dual

The Poincaré dual $Y$ corresponds to a cohomology class identifying a certain subspace of the moduli space, as explained in the previous paragraph. Its expression depends on the choice of a representative: for example, the two choices $Y_1 = \eta\delta(d\eta)$ and $Y_2 = \bar\eta\delta(d\bar\eta)$ are shown to correspond to the same class, as they both differ by exact terms to the same object. In particular, we have

$$(\eta + \bar\eta)\delta(d\eta + d\bar\eta) = \eta\delta(d\eta) - d\left[\sum_{i=1}^{\infty}\frac{\eta\bar\eta(d\bar\eta)^{i-1}}{i!}\delta^{(i)}(d\eta)\right] =$$

$$= \bar\eta\delta(d\bar\eta) + d\left[\sum_{i=1}^{\infty}\frac{\eta\bar\eta(d\eta)^{i-1}}{i!}\delta^{(i)}(d\bar\eta)\right] . \tag{5.3}$$

While the choices $Y_1$ and $Y_2$ are constructed to have zero $R$-charge, one immediately sees that *any* exact term one can add will have non-zero $R$-charge. The $d$-exact terms vanish in the $Q$-cohomology as one can directly see:

$$\int_{\mathcal{M}^{(0|2)}} [\mathrm{d}\bar\eta\mathrm{d}\eta|\mathrm{d}(\mathrm{d}\bar\eta)\mathrm{d}(\mathrm{d}\eta)] \, d\Lambda \, \langle\mathcal{V}^{(-1,0)}(-\infty)e^{iF_0}\,\mathcal{V}(0)\,e^{-iF_0}\mathcal{V}^{(0,-1)}(+\infty)\rangle =$$

$$= \pm\int_{\mathcal{M}^{(0|2)}} [\mathrm{d}\bar\eta\mathrm{d}\eta|\mathrm{d}(\mathrm{d}\bar\eta)\mathrm{d}(\mathrm{d}\eta)] \, \Lambda \, \langle\mathcal{V}^{(-1,0)}(-\infty)e^{iF_0}\,Q\mathcal{V}(0)\,e^{-iF_0}\mathcal{V}^{(0,-1)}(+\infty)\rangle , \tag{5.4}$$

where we used integration by parts (notice that $\mathcal{M}^{(0|2)}$ has no boundary) and the (co)chain map property of the picture changing operation to turn the de Rham operator on the supermoduli space to the BRST operator acting on vertices. This term then vanishes on the BRST cohomology.

## 5.2 The Middle Vertex

Let us now return to the construction of the cubic term. We start by studying the structure of the vertex operator, to be inserted at 0, in terms of its spectrum. As stated above, this vertex operator has $R$-charge 0 and picture number $-1$. Notice that the assumption on the $R$-charge (which is determined by the $R$-charge of the Poincaré dual $Y$) is compatible with that of $Q$. Thus

$$\mathcal{V}(0) = \left(\mathcal{V}_1^{(-1,0)}\delta(\gamma) + \mathcal{V}_{-1}^{(0,-1)}\delta(\bar\gamma)\right) , \tag{5.5}$$

with $\mathcal{V}_1^{(-1,0)}$ and $\mathcal{V}_{-1}^{(0,-1)}$ of $R$-charge 1 and $-1$ respectively. In the $1_\psi$ representation, the vertex $\mathcal{V}_1^{(0|-1,0)}$ reads

$$
\begin{aligned}
\mathcal{V}_1^{(-1,0)} =& A_\mu^{[1]} \psi^\mu + \left( \bar{\beta} C_{\mu\nu}^{[2]} + c\bar{\beta} Z_{\mu\nu}^{[2]} + \bar{\gamma} Z_{\mu\nu}^{*[2]} \right) \psi^\mu \psi^\nu + \\
& + \left( -\bar{\beta}\bar{\gamma} A_{\mu\nu\lambda}^{[3]} + \frac{1}{2} c\bar{\beta}^2 E_{\mu\nu\lambda}^{[3]} - \frac{1}{2}\bar{\beta}^2 D_{\mu\nu\lambda}^{[3]} \right) \psi^\mu \psi^\nu \psi^\lambda + \\
& + \left( -\frac{1}{2} c\bar{\beta}^2 \bar{\gamma} Z_{\mu\nu\lambda\rho}^{[4]} - \frac{1}{2}\bar{\gamma}\bar{\beta}^2 C_{\mu\nu\lambda\rho}^{[4]} - \frac{1}{3}\bar{\beta}^3 G_{\mu\nu\lambda\rho} + \frac{1}{3} c\bar{\beta}^3 H_{\mu\nu\lambda\rho} \right) \psi^\mu \psi^\nu \psi^\lambda \psi^\rho + \cdots ,
\end{aligned}
\tag{5.6}
$$

where "$\cdots$" corresponds to other antifields. Note that there is no room for a scalar ghost. The equations of motion and the gauge transformations are found by applying the BRST differential; the equations of motion read

$$
-i\delta A^{[3]} - id A^{[1]} - Z^{[2]} = 0 \tag{5.7a}
$$

$$
\Box A^{[1]} - i\delta Z^{[2]} = 0 \tag{5.7b}
$$

$$
\Box A^{[3]} - id Z^{[2]} - i\delta Z^{[4]} = 0 , \tag{5.7c}
$$

while the gauge transformations are

$$
\delta_{gauge} A^{[1]} = i\delta C^{[2]} , \text{ gauge for } A^{[1]} \tag{5.8a}
$$

$$
\delta_{gauge} A^{[3]} = \left( -idC^{[2]} - E^{[3]} - i\delta C^{[4]} \right) , \text{ gauge for } A^{[3]} \tag{5.8b}
$$

$$
\delta_{gauge} C^{[2]} = \frac{1}{i}\delta D^{[3]} , \text{ gauge (for gauge) for } C^{[2]} \tag{5.8c}
$$

$$
\delta_{gauge} D^{[3]} = \frac{1}{i}\delta G^{[4]} , \text{ gauge (for gauge for gauge) for } D^{[3]} \tag{5.8d}
$$

$$
\delta_{gauge} Z^{[2]} = \left( \Box C^{[2]} + \frac{1}{i}\delta E^{[3]} \right) , \text{ gauge for } Z^{[2]} \tag{5.8e}
$$

$$
\delta_{gauge} E^{[3]} = \left( \frac{1}{i}\delta H^{[4]} + \Box D^{[3]} \right) , \text{ gauge (for gauge) for } E^{[3]} . \tag{5.8f}
$$

The counting of degrees of freedom is most easily done on-shell, where $\Box \equiv 0$. Then, (5.7b) implies that $\delta Z^{[2]} = 0$ which by (5.7a) then implies Maxwell's equation for $A^{[1]}$. It might seem natural to again impose the conventional constraint $Z^{[2]} = 0$, as we did in the last section at $R$-charge 1. However, in view of the gauge invariance (5.8b,5.8e) we may also, set $A^{[3]} = 0$ (effectively transferring the components from $A^{[3]}$ to $E^{[3]}$) so that $Z^{[2]} = -id A^{[1]}$.

By following the same construction adopted in the previous sections, we want to extend the vertex by removing $1_\psi$, thus allowing for polynomials in $\bar{\psi}$ as well. We have that (5.6) is complemented by terms of the form

$$
A_{\mu\nu\rho}^{(3)} \bar{\psi}^\mu \psi^\nu \psi^\rho + c A_{\mu\nu\rho}^* \bar{\psi}^\mu \psi^\nu \psi^\rho + \cdots , \tag{5.9}
$$

plus permutations of $\psi$ and $\bar{\psi}$. Here, again, $A^{(3)}$ is of a mixed symmetry type. Analogously,

$\mathcal{V}_{-1}^{(0,-1)}$ is obtained by considering $\psi \leftrightarrow \bar{\psi}, \gamma \leftrightarrow \bar{\gamma}, \beta \leftrightarrow \bar{\beta}$. We get similarly,

$$
\begin{aligned}
\mathcal{V}_{-1}^{(0,-1)} = & A_\mu^{[1]} \bar{\psi}^\mu - \beta \bar{C}_{\mu\nu}^{[2]} \bar{\psi}^\mu \bar{\psi}^\nu - c\beta \bar{Z}_{\mu\nu}^{[2]} \bar{\psi}^\mu \bar{\psi}^\nu - \gamma \bar{Z}_{\mu\nu}^{*[2]} \bar{\psi}^\mu \bar{\psi}^\nu - \frac{1}{2} c\beta^2 \gamma Z_{\mu\nu\lambda\rho}^{[4]} \bar{\psi}^\mu \bar{\psi}^\nu \bar{\psi}^\lambda \bar{\psi}^\rho \\
& + \beta\gamma A_{\mu\nu\lambda}^{[3]} \bar{\psi}^\mu \bar{\psi}^\nu \bar{\psi}^\lambda + \gamma\beta^2 \bar{C}_{\mu\nu\lambda\rho}^{[4]} \bar{\psi}^\mu \bar{\psi}^\nu \bar{\psi}^\lambda \bar{\psi}^\rho - c\beta^2 \bar{E}_{\mu\nu\lambda}^{[3]} \bar{\psi}^\mu \bar{\psi}^\nu \bar{\psi}^\lambda \\
& + A_{\mu\nu\rho}^{(3)} \bar{\psi}^\rho \bar{\psi}^\nu \psi^\mu + c A_{\mu\nu\rho}^* \bar{\psi}^\rho \bar{\psi}^\nu \psi^\mu + \cdots .
\end{aligned}
\tag{5.10}
$$

As we did in section 3, we may identify the antisymmetric part of $A^{(3)}$ with that of $A^{[3]}$ since they gauge transform in the same way (thereby fixing a 3-form ghost $E$). Furthermore, if we set $Z^{[2]} = 0$, as described below (5.8), it is again consistent to assume that $A_{\mu\nu\rho}^{(3)}$ is completely antisymmetric.

## 5.3 Cubic Vertex

We will now obtain the cubic interaction vertex of the space-time action by integrating the pull-back over the $\mathcal{N} = 2$ moduli space. The pull-back is obtained by picture changing on the $\mathcal{N} = 2$ moduli space, given by the obvious extension from $\mathcal{N} = 1$ [23] as the finite adjoint action of

$$
F_0 = \eta(\bar{q} + b\bar{\gamma}) + \bar{\eta}(q + b\gamma) + \mathrm{d}\bar{\eta}\beta + \mathrm{d}\eta\bar{\beta} .
\tag{5.11}
$$

That still leaves us with the choice of an integration cycle, encoded in the Poincaré dual $Y$. As there is no bosonic modulus for 3 punctures (see [23]), we only need to insert one $c$-ghost to absorb the translation zero-mode (this indeed gives us a ghost number one operator inserted in 0). Let us fix the zero $R$-charge operator as $Y = \eta\delta(d\eta) + \bar{\eta}\delta(d\bar{\eta})$.

For the construction of the cubic vertex, we insert the $c$-ghost in the middle vertex operator $\mathcal{V}(0)$ given in (5.5), the picture-changing operation will have a non-trivial action on it as well. Alternatively one may insert the $c$ ghost in one of the external vertices (see [23]). However, that choice does not lead to the familiar minimal coupling in the world line supercharges below $\bar{\psi}^\mu p_\mu \mapsto \bar{\psi}^\mu(p_\mu + A_\mu)$. Our choice is thus dictated by the interest in comparing the path integral result with the deformation of the BRST charge. In particular, we have

$$
\begin{aligned}
Y e^{iF_0} c\mathcal{V}(0) e^{-iF_0} = & Y e^{iF_0} c\mathcal{V}(0)(c=0, \psi, \bar{\psi}, \gamma, \bar{\gamma}, \beta, \bar{\beta}) e^{-iF_0} = \tag{5.12}\\
= & \eta\delta(d\eta) i\bar{\eta} \Big[ -c\left\{q, \mathcal{V}_{-1}^{(0,-1)}\right\} + \gamma \mathcal{V}_{-1}^{(0,-1)}(0, \psi, \bar{\psi}, \gamma, \beta) \Big] \delta(\bar{\gamma} + id\bar{\eta}) + \\
& + \bar{\eta}\delta(d\bar{\eta}) i\eta \Big[ -c\left\{\bar{q}, \mathcal{V}_{1}^{(-1,0)}\right\} + \bar{\gamma} \mathcal{V}_{1}^{(-1,0)}(0, \psi, \bar{\psi}, \bar{\gamma}, \bar{\beta}) \Big] \delta(\gamma + id\eta) + \lambda ,
\end{aligned}
$$

where $\lambda$ is given by terms which are not proportional to $\eta\bar{\eta}$ (which then lead to zero when integrated on $\mathcal{M}^{(0|2)}$).

Let us consider explicitly the terms involving $A^{[1]}$ and $A^{(3)}$. The shift of the $c$ ghost due to picture changing produces the additional terms

$$
\bar{\gamma}(\psi^\mu A_\mu + \bar{\psi}^\mu A_{\mu\nu\rho}^{(3)} \psi^\nu \psi^\rho)\delta(\gamma) .
\tag{5.13}
$$

Performing the integral over $[\mathrm{d}\bar\eta \mathrm{d}\eta | \mathrm{d}(\mathrm{d}\bar\eta)\mathrm{d}(\mathrm{d}\eta)]$, as in (5.2) (the only terms containing these variables are in the middle vertex and in the Poincaré dual), we then find

$$\int_{\mathcal{M}^{(0|2)}} [\mathrm{d}\bar\eta \mathrm{d}\eta | \mathrm{d}(\mathrm{d}\bar\eta)\mathrm{d}(\mathrm{d}\eta)]\, Y \circ e^{iF_0}\, c\mathcal{V}(0)\, e^{-iF_{s_0}} = \tag{5.14}$$
$$-2i(\mathrm{d}A)_{\mu\nu} : \bar\psi^\mu \psi^\nu : +\{A^\mu, p_\mu\} + \bar\gamma(\psi^\mu A_\mu + \bar\psi^\mu A^{(3)}_{\mu\nu\rho} \psi^\nu \psi^\rho) + \gamma(\bar\psi^\mu A_\mu - \psi^\mu A^{(3)}_{\mu\nu\rho} \bar\psi^\nu \bar\psi^\rho)$$
$$\equiv c\Delta H(A^{[1]}) + \bar\gamma \Delta q(A^{[1]}, A^{[3]}) + \gamma \Delta \bar q(A^{[1]}, A^{[3]})\,,$$

where the last line makes contact with the deformed BRST operator in section 2. Here, we suppressed ghosts and antifields as well as contributions that vanish when inserted in the 3-point correlation function. Note that $\Delta H(A^{[1]})$ is independent of $A^{[3]}$ and $Z^{[2]}$ with our choice of the Poincaré dual. This will be important for comparison with the deformation problem in section 2.

We are now ready to evaluate the cubic interaction. For this we insert (5.14) into the 3-point function (5.2),

$$S^{(3)} = \langle \mathcal{V}^{(-1,0)}(-\infty) \left[ c\Delta H(A^{[1]}) + \bar\gamma \Delta q(A^{[1]}, A^{[3]}) + \gamma \Delta \bar q(A^{[1]}, A^{[3]}) \right] \mathcal{V}^{(0,-1)}(\infty) \rangle\,. \tag{5.15}$$

where $\mathcal{V}^{(0,-1)}(-\infty)$ and $\mathcal{V}^{(-1,0)}(\infty)$ are given by the completion of (3.4) by higher form fields, as described in section 3. As such, $S^{(3)}$ describes the interaction between p-form fields (assuming the conventional constraint $B^{[2]} = 0$, in sec. 3) in analogy with the $\mathcal{N} = 1$ world line in [23].

## 5.4 Relation to the Deformation Problem

To make contact with the Fock space description reviewed in section 2, we may project the vertex operators onto Fock states by re-inserting the ground state $1_\psi$. That is

$$\mathcal{V}^{(0,-1)}(\infty) \overset{\pi}{\mapsto} \mathcal{V}^{(0,-1)}(\infty)1_\psi = (A_\mu \psi^\mu + \beta C + c\beta iB)\, \delta(\bar\gamma)\,. \tag{5.16}$$

Then (5.15) reduces to

$$\langle A_\mu \bar\psi^\mu \delta(\gamma) c \left( -2i(\mathrm{d}A)_{\mu\nu} : \bar\psi^\mu \psi^\nu : +\{A^\mu, p_\mu\} \right) A_\mu \psi^\mu \delta(\bar\gamma) \rangle$$
$$= \frac{1}{2} A^\mu (-i(\mathrm{d}A)_{\mu\nu} + \{p^\alpha, A_\alpha\} g_{\mu\nu}) A^\nu\,, \tag{5.17}$$

with further contributions of ghosts and auxiliary fields. Combining this with the quadratic contribution from (4.4) we have

$$S^{(2)} + S^{(3)} = -\langle A_\mu \bar\psi^\mu \delta(\gamma)\cdot$$
$$\left[ -c(p^2 + \{A^\mu, p_\mu\}\ -2i(\mathrm{d}A)_{\mu\nu} : \bar\psi^\mu \psi^\nu :) + \gamma\bar\psi\cdot(p+A) + \bar\gamma\psi\cdot(p+A) + b\bar\gamma\gamma \right] A_\mu \psi^\mu \delta(\bar\gamma) \rangle$$
$$\equiv -\langle A_\mu \bar\psi^\mu \delta(\gamma)\, Q(A)\, A_\mu \psi^\mu \delta(\bar\gamma) \rangle \tag{5.18}$$

and additional terms involving auxiliary fields, ghost and anti fields. Comparing $Q(A)$ obtained in this way with (2.9) determined by the deformation problem reviewed in section 2 and recalling that $(\mathrm{d}A)_{\mu\nu} = \frac{1}{2}(\partial_\mu A_\nu - \partial_\nu A_\mu)$ it appears that there is a mismatch of a factor

2 in the deformation of $H$ in the two approaches. However, there is a subtlety in comparing the evaluation of $Q$ in the path-integral and to the action of $Q$ on the Fock module: in the path-integral evaluation we use the Wick contraction as in section 2, $\langle \bar{\psi}^\mu(t)\psi^\nu(t')\rangle = \frac{1}{2}\mathrm{sgn}(t-t')g^{\mu\nu}$ (which is the origin of the $1/2$ in the second line of (5.17)), while in the operator formulation on the Fock space we have $\bar{\psi}^\mu\psi^\nu|1\rangle = g^{\mu\nu}$. Thus, to compare the two formulations, we should rescale the $(\bar{\psi},\psi)$-algebra in the operator formulation by $\frac{1}{2}$, which, in turn, amounts to setting $\kappa = \frac{1}{2}$ in (2.9). Taking this into account, we then see that the cubic term in the path-integral quantization precisely reproduces the deformation problem to this order.

## 6 The Quartic Interaction

In this section, we analyse the structure of the quartic term emerging from the prescriptions described in the previous sections. In particular, we show how the presence of the Poincaré duals allows us to derive the expected form of the quartic interaction obtained in the $\mathcal{N}=1$ model.

### 6.1 Associator in $\mathcal{N}=2$

In the previous paragraphs, we have shown that in order to find the desired deformation of the BRST charge $Q$ (see eq. (5.18)), we have to define a product of vertex operators with the insertion of picture changing operators and Poincaré duals. By following the prescription given in (5.14), this results in a binary product defined as

$$m_2(\mathcal{V}_1,\mathcal{V}_2) := \int_{\mathcal{M}^{(0|2)}}[\mathrm{d}\bar{\eta}\mathrm{d}\eta|\mathrm{d}(\mathrm{d}\bar{\eta})\mathrm{d}(\mathrm{d}\eta)]\, Y \circ e^{iF_0}\, c\mathcal{V}_1\, e^{-iF_0}\, \mathcal{V}_2\,. \tag{6.1}$$

In particular, we can rewrite the cubic term of the action as

$$S^{(3)} = \left\langle \mathcal{V}^{(-1,0)}(-\infty)m_2\left(\mathcal{V}(0),\mathcal{V}^{(0,-1)}(\infty)\right)\right\rangle\,. \tag{6.2}$$

The gauge structure is governed by the $A_\infty$ structure[8] and, in particular, the non-associativity of $m_2$ determines the failure of the action to be gauge invariant. This is compensated for by the introduction of a suitable quartic term, as we show in the next subsection.

The associator of $m_2$ is defined as

$$Ass(\mathcal{V}_1,\mathcal{V}_2,\mathcal{V}_3) := m_2\left(m_2\left(\mathcal{V}_1,\mathcal{V}_2\right),\mathcal{V}_3\right) - m_2\left(\mathcal{V}_1,m_2\left(\mathcal{V}_2,\mathcal{V}_3\right)\right)\,, \tag{6.3}$$

and at the level of the path integral, it leads to

$$\int_{\mathcal{M}^{(0|4)}}[\mathrm{d}\bar{\eta}^2\mathrm{d}\eta^2|\mathrm{d}(\mathrm{d}\bar{\eta})^2\mathrm{d}(\mathrm{d}\eta)^2]\,\left\langle Y_0 Y_\epsilon \mathcal{V}(-\infty)e^{iF_0}ce^{iF_\epsilon}c\mathcal{V}(0)e^{-iF_\epsilon}\mathcal{V}(\epsilon)e^{-iF_0}\mathcal{V}(\infty)\right\rangle +$$
$$- \left\langle Y_0 Y_\epsilon \mathcal{V}(-\infty)e^{iF_0}c\mathcal{V}(0)e^{-iF_0}e^{iF_\epsilon}c\mathcal{V}(\epsilon)e^{-iF_\epsilon}\mathcal{V}(\infty)\right\rangle\,, \tag{6.4}$$

where $Y_0$ and $Y_\epsilon$ are two Poincaré duals, coming with the insertion of the operators in the punctures in $0$ and $\epsilon$, respectively. Here, indeed, one has four odd moduli to integrate over, two of them are trivialised by the insertion of the Poincaré duals in order to reproduce the $\mathcal{N}=1$ model.

---

[8]Instead of an $A_\infty$ complex, we should rather talk about the cyclic complex, as the $A_\infty$ relations are verified once inserted in the inner product given by the path integration.

## 6.2 The Quartic Term

The structure of the associator in (6.4) determines the quartic vertex, which thus reads (we omit the integration variable)

$$
\begin{aligned}
S^{(4)} = \int_{\mathcal{M}^{(1|4)}} Y_0 Y_\epsilon &\left\langle \mathcal{V}(-\infty)e^{iF_0}e^{i(G_\tau+F_\tau)}\mathcal{V}(0)e^{-i(G_\tau+F_\tau)}\mathcal{V}(\epsilon)e^{-iF_0}\mathcal{V}(\infty)\right\rangle + \\
- Y_0 Y_\epsilon &\left\langle \mathcal{V}(-\infty)e^{iF_0}\mathcal{V}(0)e^{-iF_0}e^{i(G_\tau+F_\tau)}\mathcal{V}(\epsilon)e^{-i(G_\tau+F_\tau)}\mathcal{V}(\infty)\right\rangle ,
\end{aligned} \tag{6.5}
$$

where

$$
G_\tau(\tau) = g_\tau(\tau)H + \mathfrak{g}_\tau(\tau)b , \qquad \mathfrak{g}_\tau(s) = \frac{\partial g(s)}{\partial \tau}\mathrm{d}\tau . \tag{6.6}
$$

and $g_\tau = \tau\Phi(s)$, where $\Phi(s)$ is a differentiable function that takes the value 1 in the interval $[0, \epsilon]$ (see also [23]). The introduction of a fourth puncture results in a (1|4)-dimensional supermoduli space, in contrast to the (1|2)-dimensional case for the $\mathcal{N} = 1$ model. Indeed, the insertion of the fourth puncture increases the dimension by (1|2). The even modulus represents the "length of the stub", or, analogously, the free parameter encoding the distance between the two middle punctures. The two extra odd moduli represent the freedom of the fourth vertex in the two algebraic directions (roughly speaking, if these were geometric directions, these two moduli would encode the freedom in moving the new puncture).

One can check that this quartic term compensates for the associator of the previous section. Considering the path integration (i.e., the cyclic structure), it defines a quaternary operation that we denote by $C_4$. By acting with the BRST operator $Q$ and by using the fact that the picture-changing operation is a (co)chain map between the BRST complex and the de Rham complex[9] on $\mathcal{M}^{(1|4)}$[10], we get

$$
\begin{aligned}
C_4 \circ (Q \otimes Id^{\otimes 3} + Id \otimes Q \otimes Id^{\otimes 2} + Id^{\otimes 2} \otimes Q \otimes Id + Id^{\otimes 3} \otimes Q) = \\
= (d\eta_\tau \partial_{\eta_\tau} + d\bar{\eta}_\tau \partial_{\bar{\eta}_\tau} + d\eta_0 \partial_{\eta_0} + d\bar{\eta}_0 \partial_{\bar{\eta}_0} + d\tau \partial_\tau)C_4 .
\end{aligned} \tag{6.7}
$$

With respect to the integration over $\mathcal{M}^{(1|4)}$, the only relevant term comes from $d\tau \partial_\tau$ and its contribution to the $\epsilon$ boundary (recall that the bosonic modulus $\tau$ describes the distance between the puncture in 0 and the puncture in $\epsilon$). This leads to

$$
\begin{aligned}
\int_{\mathcal{M}^{(0|4)}} &\left\langle Y_0 Y_\epsilon \mathcal{V}(-\infty)e^{iF_0}e^{iF_\epsilon}\mathcal{V}(0)e^{-iF_\epsilon}\mathcal{V}(\epsilon)e^{-iF_0}\mathcal{V}(\infty)\right\rangle + \\
- &\left\langle Y_0 Y_\epsilon \mathcal{V}(-\infty)e^{iF_0}\mathcal{V}(0)e^{-iF_0}e^{iF_\epsilon}\mathcal{V}(\epsilon)e^{-iF_\epsilon}\mathcal{V}(\infty)\right\rangle ,
\end{aligned} \tag{6.8}
$$

which indeed coincides with (6.4).

---

[9]With an abuse of notations, we call here "de Rham complex" the direct sum of complexes in different pictures on a supermanifold. This is in line with the realisation of picture via formal Dirac delta distributions (see, e.g., [24]).

[10]Analogously, one could use the two Poincaré duals to realise picture changing operators as (co)chain maps from the BRST complex and the de Rham complex on the reduced space $\mathcal{M}^{(1|2)}$.

Let us consider the insertion of Poincaré duals with the same structure as described in section 5:

$$Y_0 Y_\epsilon = \left(\eta_0 \delta(d\eta_0) + \bar\eta_0 \delta(d\bar\eta_0)\right)\left(\eta_\epsilon \delta(d\eta_\epsilon) + \bar\eta_\epsilon \delta(d\bar\eta_\epsilon)\right) \ . \tag{6.9}$$

We start from the first line of (6.5) and consider the term proportional to $\eta_0\delta(d\eta_0)\eta_\epsilon\delta(d\eta_\epsilon)$ in (6.9) using (5.12). This results in ( denoting the position as subscripts )

$$\eta_0\delta(d\eta_0)\eta_\epsilon\delta(d\eta_\epsilon)\mathcal{V}_{-\infty}(1+i\bar\eta_0 q)(1+i\bar\eta_\epsilon q)e^{igH}(c+i\bar\eta_0\gamma+i\bar\eta_\epsilon\gamma+i\mathfrak{g})\cdot \tag{6.10}$$
$$\mathcal{V}_0(0,\gamma,\bar\gamma+id\bar\eta_0+id\bar\eta_\epsilon)e^{-igH}(1-i\bar\eta_\epsilon q)(c+i\bar\eta_0\gamma)\mathcal{V}_\epsilon(0,\gamma,\bar\gamma+id\bar\eta_0)(1-i\bar\eta_0 q)\mathcal{V}_\infty \ .$$

In order to have a non-trivial integration on $\mathcal{M}^{(1|4)}$, we need to select the term linear in $\mathfrak{g}$, since it is the only term containing $d\tau$). The $c$ ghost in the third term of the second line contributes as (we omit the arguments of the vertices)

$$\eta_0\delta(d\eta_0)\eta_\epsilon\delta(d\eta_\epsilon)\mathcal{V}_{-\infty}(1+i\bar\eta_0 q)(1+i\bar\eta_\epsilon q)e^{igH}i\mathfrak{g}\mathcal{V}_0 e^{-igH}(1-i\bar\eta_\epsilon q)c\mathcal{V}_\epsilon(1-i\bar\eta_0 q)\mathcal{V}_\infty \ . \tag{6.11}$$

For a non-zero Berezin integration, we then need $\bar\eta_0\bar\eta_\epsilon$ multiplying $q$. This gives rise to the structure

$$-\eta_0\delta(d\eta_0)\eta_\epsilon\delta(d\eta_\epsilon)\bar\eta_0\bar\eta_\epsilon i\mathfrak{g}c\mathcal{V}_{-\infty}[q,[q,e^{igH}\mathcal{V}_0 e^{-igH}]\mathcal{V}_\epsilon]\mathcal{V}_\infty \ . \tag{6.12}$$

On the other hand, if we assume that the ghost $c$ comes from either one of the vertices in $\pm\infty$, this picks $i\bar\eta_0\gamma$ from the third term in the second line of (6.10) leading to

$$\eta_0\delta(d\eta_0)\eta_\epsilon\delta(d\eta_\epsilon)\gamma\bar\eta_0\bar\eta_\epsilon i\mathfrak{g}\mathcal{V}_{-\infty}[q,e^{igH}\mathcal{V}_0 e^{-igH}]\mathcal{V}_\epsilon\mathcal{V}_\infty \ . \tag{6.13}$$

All the vertices are now evaluated in $c=0$. We are only interested in the $\epsilon\to 0$ limit: the only non-zero contributions come from the boundary term in the $\tau$ integration resulting from $[q,\bar q]\sim H$, which gives a total $\tau$ derivative. As one can directly see from (6.12) and (6.13), this structure can only be recovered from those terms in the Poincaré duals in (6.9) combining $\eta$ and $\bar\eta$, thus immediately showing that both (6.12) and (6.13) are zero.

Next we consider the first line of (6.5), proportional to $\eta_0\delta(d\eta_0)\bar\eta_\epsilon\delta(d\bar\eta_\epsilon)$ of the Poincaré dual. This gives

$$\eta_0\delta(d\eta_0)\bar\eta_\epsilon\delta(d\bar\eta_\epsilon)\mathcal{V}_{-\infty}(1+i\bar\eta_0 q)(1+i\eta_\epsilon\bar q)e^{igH}(c+i\bar\eta_0\gamma+i\eta_\epsilon\bar\gamma+i\mathfrak{g})\cdot \tag{6.14}$$
$$\mathcal{V}_0(0,\gamma+id\eta_\epsilon,\bar\gamma+id\bar\eta_0)e^{-igH}(1-i\eta_\epsilon\bar q)(c+i\bar\eta_0\gamma)\mathcal{V}_\epsilon(0,\gamma,\bar\gamma+id\bar\eta_0)(1-i\bar\eta_0 q)\mathcal{V}_\infty \ .$$

With analogous arguments as above, we now obtain

$$-\eta_0\delta(d\eta_0)\bar\eta_\epsilon\delta(d\bar\eta_\epsilon)\bar\eta_0\eta_\epsilon i\mathfrak{g}c\mathcal{V}_{-\infty}[q,[\bar q,e^{igH}\mathcal{V}_0 e^{-igH}]\mathcal{V}_\epsilon]\mathcal{V}_\infty \ , \tag{6.15}$$

with the combination $[q,[\bar q,\cdot]]$ now leading to a non-trivial contribution after $\tau$ integration.

Let us first focus on the terms containing $A^{[1]}$ only. After integration over $\tau$ and $d\tau$ we are left with

$$\eta_0\delta(d\eta_0)\bar\eta_\epsilon\delta(d\bar\eta_\epsilon)\bar\eta_0\eta_\epsilon\mathcal{V}_{-\infty}\,(\gamma)\,(A_\nu\bar\psi^\nu)(0)\delta(\gamma+id\eta_\epsilon)c(A_\rho\psi^\rho)(\epsilon)\delta(\bar\gamma+id\bar\eta_0)\mathcal{V}_\infty$$

Similarly the contribution proportional to $\bar{\eta}_0 \delta(d\bar{\eta}_0) \eta_\epsilon \delta(d\eta_\epsilon)$ of the Poincaré dual gives

$$\bar{\eta}_0 \delta(d\bar{\eta}_0) \eta_\epsilon \delta(d\eta_\epsilon) \bar{\eta}_\epsilon \eta_0 \mathcal{V}_{-\infty} (\gamma) (A_\nu \psi^\nu)(0) \delta(\bar{\gamma} + id\bar{\eta}_\epsilon) c (A_\rho \bar{\psi}^\rho)(\epsilon) \delta(\gamma + id\eta_0) \mathcal{V}_\infty \,.$$

Integrating over $\mathcal{M}$ and keeping those terms leading to non-vanishing contributions in the path integral (6.5) finally reduces to

$$\langle \mathcal{V}_{-\infty} \left( A^2 + [A_\mu, A_\nu] : \bar{\psi}^\mu \psi^\nu : \right) \mathcal{V}_\infty \rangle \,. \tag{6.16}$$

Combining this with (5.18) we end up with

$$S^{(2)} + S^{(3)} + S^{(4)} = -\langle A_\mu \bar{\psi}^\mu \delta(\gamma) \, Q(A) \, A_\mu \psi^\mu \delta(\bar{\gamma}) \rangle \,, \tag{6.17}$$

where $Q(A)$ agrees with the complete deformed BRST operator in (2.9).

Finally, we consider the terms with $A^{(3)}$ in the middle vertices. We will set $Z^{[2]} = 0$ which allows us to assume that $A^{(3)}$ is totally antisymmetric, i.e. $A^{(3)} = A^{[3]}$. As in (5.16), we project the external vertex operators with $1_\psi$ and focus on the $Y_0 \bar{Y}_\epsilon$ sector. Proceeding as above, starting from (6.15) and integrating the total derivative over the even modulus, we find

$$A_\mu \bar{\psi}^\mu \delta(\gamma_{-\infty}) \left( A_{\mu_0} \psi^{\mu_0} + A_{\mu_0 \nu_0 \rho_0} \psi^{\mu_0} \psi^{\nu_0} \bar{\psi}^{\rho_0} + \text{perm} - \bar{\beta}\bar{\gamma} A_{\mu_0 \nu_0 \rho_0} \psi^{\mu_0} \psi^{\nu_0} \psi^{\rho_0} \right) \delta(\gamma_\epsilon + id\eta_\epsilon) \cdot \tag{6.18}$$

$$\left( A_{\mu_\epsilon} \bar{\psi}^{\mu_\epsilon} - A_{\mu_\epsilon \nu_\epsilon \rho_\epsilon} \psi^{\mu_\epsilon} \bar{\psi}^{\nu_\epsilon} \bar{\psi}^{\rho_\epsilon} + \text{perm} + \beta\gamma A_{\mu_\epsilon \nu_\epsilon \rho_\epsilon} \bar{\psi}^{\mu_\epsilon} \bar{\psi}^{\nu_\epsilon} \bar{\psi}^{\rho_\epsilon} \right) \delta(\bar{\gamma}_0 + id\bar{\eta}_0) A_\nu \psi^\nu \delta(\bar{\gamma}_\infty) \,.$$

where perm stands for permutations of $\psi$ and $\bar{\psi}$ as in (5.9). The only terms linear in $A^{[3]}$ are:

$$A_\mu \bar{\psi}^\mu A_{\mu_0 \nu_0 \rho_0} \psi^{\mu_0} \psi^{\nu_0} \bar{\psi}^{\rho_0} A_{\mu_\epsilon} \bar{\psi}^{\mu_\epsilon} A_\nu \psi^\nu - A_\mu \bar{\psi}^\mu A_{\mu_0} \psi^{\mu_0} A_{\mu_\epsilon \nu_\epsilon \rho_\epsilon} \psi^{\mu_\epsilon} \bar{\psi}^{\nu_\epsilon} \bar{\psi}^{\rho_\epsilon} A_\nu \psi^\nu \,, \tag{6.19}$$

Contracting the $\psi$'s, we end up with

$$-\frac{1}{4} A^{\mu_0} A_{\mu_0 \nu_0 \rho_0} A^{\nu_0} A^{\rho_0} + \frac{1}{4} A^{\mu_\epsilon} A^{\nu_\epsilon} A_{\mu_\epsilon \nu_\epsilon \rho_\epsilon} A^{\rho_\epsilon} \tag{6.20}$$

which vanishes using the cyclicity of the trace. An analogous argument holds for the terms coming from the $\bar{Y}_0 Y_\epsilon$ sector of the Poincaré dual.

Let us now consider the quadratic terms in $A^{[3]}$, again in the $Y_0 \bar{Y}_\epsilon$ sector. Working out the contractions of the ghosts, we have

$$A_\mu \bar{\psi}^\mu \left( -A_{\mu_0 \nu_0 \rho_0} \psi^{\mu_0} \psi^{\nu_0} \bar{\psi}^{\rho_0} A_{\mu_\epsilon \nu_\epsilon \rho_\epsilon} \psi^{\mu_\epsilon} \bar{\psi}^{\nu_\epsilon} \bar{\psi}^{\rho_\epsilon} + \text{perm} \right. \tag{6.21}$$

$$\left. + A_{\mu_0 \nu_0 \rho_0} \psi^{\mu_0} \psi^{\nu_0} \psi^{\rho_0} A_{\mu_\epsilon \nu_\epsilon \rho_\epsilon} \bar{\psi}^{\mu_\epsilon} \bar{\psi}^{\nu_\epsilon} \bar{\psi}^{\rho_\epsilon} \right) A_\nu \psi^\nu \,.$$

It then follows that all possible contractions add up to zero due to the relative sign and the cyclicity of the trace. This thus shows that when considering the projection on the external states, the higher form field $A^{[3]}$ decouples, in analogy to what we found for the cubic term.

### 6.3 No Higher Terms

The next step of the construction should be the five-points function. This corresponds to inserting five vertex operators on the world line, corresponding to a $(2|6)$ dimensional supermoduli space, the two even moduli corresponding to the "length" of the two stubs or, analogously, to the two distances between the three middle punctures. Nonetheless, we can show that this correlator (as well as all the higher ones) vanishes in the zero-length limit of the stubs. This would not be true if we were considering a full $\mathcal{N} = 2$ model, i.e., without the insertion of Poincaré duals restricting to a subspace.

Let us consider one of the possible expressions that could define the quintic vertex, with the insertion of three Poincaré duals, that is, one for every middle puncture as described in the previous sections:

$$S^{(5)} = \int_{\mathcal{M}^{(2|6)}} Y_0 Y_{\epsilon_1} Y_{\epsilon_2} \left\langle \mathcal{V}(-\infty) e^{iF_0} e^{i(G_{\tau_1}+F_{\tau_1})} e^{i(G_{\tau_2}+F_{\tau_2})} \mathcal{V}(0) e^{-i(G_{\tau_2}+F_{\tau_2})} \right.$$
$$\left. \mathcal{V}(\epsilon_1) e^{-i(G_{\tau_1}+F_{\tau_1})} \mathcal{V}(\epsilon_2) e^{-iF_0} \mathcal{V}(\infty) + \dots \right\rangle , \tag{6.22}$$

where "..." indicates other possible terms of the quintic correlator. When considering the $\epsilon_1, \epsilon_2 \to 0$ limit, only the boundary terms will give a non-trivial contribution, as for the quartic interaction term, where they emerged from the double supercharge coming from the two picture changing operations resulting in the operator $H$. In the present case of (6.22), we would need this operation twice. However, the prescription of introducing three Poincaré duals, reflecting to the fact that we are picture changing three times, will lead to at most three supercharges, that is, at most one translation operator $H$. Thus, any term of this form will vanish in the aforementioned limit, confirming the absence of higher terms in the action.

## 7 Discussion

To summarize, in this paper we have shown that the perturbative path-integral quantization of the $\mathcal{N} = 2$ spinning particle does indeed give an off-shell description of Yang-Mills theory upon a suitable restriction of the supermoduli space. This restriction is described by a choice of a Poincaré dual. This choice is not unique, but we argued that different choices give equivalent on-shell amplitudes and therefore should correspond to field redefinitions. Unlike in string theory, the (vertex) operator-state map is surjective only, rather than an isomorphism. Therefore, embedding of states in the operator algebra involves some choices. We formulated this embedding as a quasi-isomorphism between the relevant cohomologies at a given picture and $R$-charge. However, on the level of the target space action, this isomorphism becomes manifest only after eliminating a number of auxiliary fields.

Turning on interactions requires a further, non-standard formulation of the BV-multiplet, due to R-charge conservation of the $\mathcal{N} = 2$ world line. After establishing equivalence, we embed the latter in the operator algebra and show that integration over supermoduli space $\mathcal{M}$ maps this multiplet into a deformation of the BRST operator acting on the perturbative Fock space. This is summarized in the following diagrams:

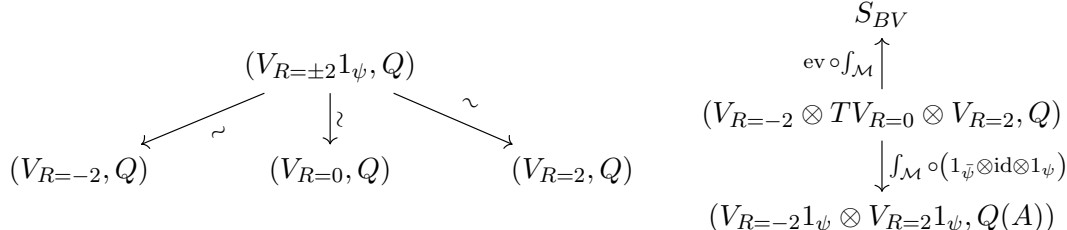

Here $ev$ is realized by the path integral, while $1_\psi$ and $1_{\bar\psi}$ give the projection on the Fock space. The deformation $Q(A)$ was previously motivated by demanding that nilpotency implies the non-linear Yang-Mills equation [16]. Here, we derive it from the path integral as a result of picture changing. In the process, we also derive a non-linear space-time action directly from the world line theory, which was previously unavailable, primarily due to the absence of trace on the relevant operator algebra [13]. The path integral provides such a trace in a larger algebra, at the price of generating higher form fields. The latter feature is a world-line realization of the familiar fact in the world-sheet approach of string theory that the operator product expansion does not commute with level truncation.

This larger vertex algebra looks somewhat like a familiar form of the duality invariant Yang-Mills theory realized on the $\mathcal{N}=1$ world line [23]. Indeed, one can embed the latter in $\mathcal{N}=2$ with a suitable projection in the ghost sector. Perhaps this can lead to some insight into what the non-projected $\mathcal{N}=2$ theory describes. Conversely, the presence of $\mathcal{N}=2$ world-line SUSY enabled us to project out higher form fields in the external state, which was not possible in $\mathcal{N}=1$ (akin to the inconsistency of level truncation in string theory). This was instrumental in order to recover the deformation problem described in section 2. A further non-trivial observation is that with our choice of $Y$, the middle vertex did not give rise to interactions with higher form fields either. This is crucial for the consistency of the deformation problem. One may note some similarity with the presence of enhanced global symmetries in certain string backgrounds (e.g. [28–30]) except that the decoupling holds for an arbitrary space-time profile. Perhaps this could help in relation to level truncation in string field theory.

The space-time action obtained through our procedure depends on the choice of the Poincaré dual, $Y$ on the moduli space. We argue that different choices amount to field redefinitions. An interesting observation is that there is a choice of $Y$ for which there is no quartic interaction. On the other hand, with this choice, the cubic term does not have a familiar form Yang-Mills theory and furthermore involves higher form auxiliary fields, in agreement with general expectations.

A natural extension of the present formulation is to remove the Poincaré dual $Y$ introduced here. This should lead to a higher derivative gauge theory whose gauge invariance is guaranteed by the pull-back to the supermoduli space. Given that vertex operator algebra contains mixed symmetry tensors, in general (as opposed to the Fock space projection), gravitational interaction will naturally appear (as is expected for $\mathcal{N}=2$), possibly in a duality invariant formulation. Embedding that cohomology in $\mathcal{N}=4$, one may be able to derive the deformed BRST operator whose nilpotency was shown to reproduce the non-linear equation for the massless fields in string theory [17]. An interesting question regards

the role of the cosmological constant, which was unconstrained in the deformation problem.

In view of a possible extension to string theory, one faces the complication of representing geometric objects such as $\delta(\gamma)$ as well as Poincaré duals on the world-sheet conformal field theory. The bosonized $\beta - \gamma$ system obscured the connection to moduli space. Here, the embedding of the $\mathcal{N} = 1$ string in the $\mathcal{N} = 2$ world sheet theory described in [26, 27] could suggest a way forward.

## Acknowledgments

We would like to thank Nathan Berkovitz, Alberto Cattaneo, Martin Cederwall, Chris Hull, Ezra Getztler and Ashoke Sen for helpful discussions. Part of this work was done while I.S. took part in the program "Cohomological aspects of Quantum Field Theory" 2025, at Mittag-Leffler Institute in Stockholm, supported by the Swedish Research Council under grant no. 2021-06594. This work was supported in part by the Excellence Cluster Origins of the DFG under Germany's Excellence Strategy EXC-2094 390783311.

## A   No $B^{(2)}$

In this section, we want to give a different prescription for the construction of the state-operator map $\iota$ of Section 3. In particular, we want to avoid the use of the 2-tensor $B_{\mu\nu}$, in analogy to setting it to zero as a constraint (see Section 3).

Let us define

$$
\iota\left(A_\mu\psi^\mu 1_\psi\right) = A_\mu\psi^\mu - \beta\gamma A_\mu\bar{\psi}^\mu + A_{\alpha\mu\nu}(\psi\psi\bar{\psi})^{\alpha\mu\nu} - \beta\gamma A_{\alpha\mu\nu}(\psi\bar{\psi}\bar{\psi})^{\alpha\mu\nu} + \frac{1}{2}\beta^2\gamma^2 A_{\alpha\mu\nu}(\bar{\psi}\bar{\psi}\bar{\psi})^{\alpha\mu\nu}
$$
$$
+ c\beta B_{\alpha\beta\mu\nu}\left(\psi\psi\bar{\psi}\bar{\psi}\right)^{\alpha\beta\mu\nu} - \frac{1}{2}c\beta^2\gamma B_{\alpha\beta\mu\nu}\left(\psi\bar{\psi}\bar{\psi}\bar{\psi}\right)^{\alpha\beta\mu\nu} + \frac{1}{3!}c\beta^3\gamma^2 B_{\alpha\beta\mu\nu}\left(\bar{\psi}\bar{\psi}\bar{\psi}\bar{\psi}\right)^{\alpha\beta\mu\nu} \ ,
$$
$$
\text{(A.1)}
$$

where we assume all fields to be totally antisymmetric. We want to verify under which conditions on the fields $A_{\mu\nu\rho}$ and $B_{\mu\nu\rho\sigma}$ the map $\iota$ is a cochain map, i.e., $\iota \circ Q = Q \circ \iota$. On the left-hand side, we have (for the moment, we assume that the $c$ and $\gamma$ sectors are embedded "trivially")

$$
QA_\mu\psi^\mu 1_\psi = c\Box A_\mu\psi^\mu 1_\psi + \gamma\frac{1}{i}\delta A 1_\psi \ , \ \iota\left(QA_\mu\psi^\mu 1_\psi\right) = c\Box A_\mu\psi^\mu + \gamma\frac{1}{i}\delta A \ . \qquad \text{(A.2)}
$$

On the right-hand side, we have

$$Q\left(\iota A_\mu \psi^\mu 1_\psi\right) = c\left(\Box A_\mu \psi^\nu + \Box A_{\alpha\mu\nu}(\psi\psi\bar\psi)^{\alpha\mu\nu} + qB_{\alpha\beta\mu\nu}\left(\psi\psi\bar\psi\bar\psi\right)^{\alpha\beta\mu\nu}\right) +$$
$$+ \gamma\left(\frac{1}{i}dA^{(1)} + \frac{1}{i}\delta A^{(1)} + \bar{q}A_{\alpha\mu\nu}(\psi\psi\bar\psi)^{\alpha\mu\nu} + qA_{\alpha\mu\nu}(\psi\bar\psi\bar\psi)^{\alpha\mu\nu} - B_{\alpha\beta\mu\nu}\left(\psi\psi\bar\psi\bar\psi\right)^{\alpha\beta\mu\nu}\right) +$$
$$+ c\beta\gamma\left(-\Box A_\mu\bar\psi^\mu - \Box A_{\alpha\mu\nu}(\psi\bar\psi\bar\psi)^{\alpha\mu\nu} - \bar{q}B_{\alpha\beta\mu\nu}\left(\psi\psi\bar\psi\bar\psi\right)^{\alpha\beta\mu\nu} - qB_{\alpha\beta\mu\nu}\left(\psi\bar\psi\bar\psi\bar\psi\right)^{\alpha\beta\mu\nu}\right) +$$
$$+ \beta\gamma^2\left(\frac{1}{i}dA^{(1)} - qA_{\alpha\mu\nu}(\bar\psi\bar\psi\bar\psi)^{\alpha\mu\nu} - \bar{q}A_{\alpha\mu\nu}(\psi\bar\psi\bar\psi)^{\alpha\mu\nu} + B_{\alpha\beta\mu\nu}\left(\psi\bar\psi\bar\psi\bar\psi\right)^{\alpha\beta\mu\nu}\right) +$$
$$+ c\beta^2\gamma^2\left(+\frac{1}{2}\Box A_{\alpha\mu\nu}(\bar\psi\bar\psi\bar\psi)^{\alpha\mu\nu} + \frac{1}{2}\bar{q}B_{\alpha\beta\mu\nu}\left(\psi\bar\psi\bar\psi\bar\psi\right)^{\alpha\beta\mu\nu} + \frac{1}{2}qB_{\alpha\beta\mu\nu}\left(\bar\psi\bar\psi\bar\psi\bar\psi\right)^{\alpha\beta\mu\nu}\right) +$$
$$+ \beta^2\gamma^3\left(\frac{1}{2}\bar{q}A_{\alpha\mu\nu}(\bar\psi\bar\psi\bar\psi)^{\alpha\mu\nu} - \frac{1}{2}B_{\alpha\beta\mu\nu}\left(\bar\psi\bar\psi\bar\psi\bar\psi\right)^{\alpha\beta\mu\nu}\right) . \tag{A.3}$$

Notice that, by assuming complete antisymmetry, we can rewrite the following expressions as

$$\bar{q}A_{\alpha\mu\nu}(\psi\psi\bar\psi)^{\alpha\mu\nu} + qA_{\alpha\mu\nu}(\psi\bar\psi\bar\psi)^{\alpha\mu\nu} = \frac{1}{i}dA^{(3)} + \frac{1}{i}\delta A^{(3)} , \tag{A.4}$$

$$\bar{q}B_{\mu\nu\rho\sigma}\left(\psi\psi\bar\psi\bar\psi\right)^{\mu\nu\rho\sigma} + qB_{\mu\nu\rho\sigma}\left(\psi\bar\psi\bar\psi\bar\psi\right)^{\mu\nu\rho\sigma} = \frac{1}{i}\delta B^{(4)}. \tag{A.5}$$

In the $\gamma$-sector, by comparing with (A.2), we find the following constraints

$$B^{(4)} = \frac{1}{i}dA^{(3)} , \quad \delta A^{(3)} + dA^{(1)} = 0 . \tag{A.6}$$

They imply

$$\Box A^{(3)} = (d\delta + \delta d)A^{(3)} = \delta dA^{(3)} + d(-dA^{(1)}) = \delta dA^{(3)} = i\delta B^{(4)} . \tag{A.7}$$

By imposing the conditions (A.6), we see that the last three lines of (A.3) vanish and the $c$ and $\gamma$ sectors agree with (A.2). The $c\beta\gamma$ sector though does not agree with (A.2), thus suggesting a modification of the definition of $\iota$ in the $c$ sector, as

$$cA_\mu^*\psi^\mu 1_\psi \overset{\iota}{\mapsto} cA_\mu^*\psi^\mu - c\beta\gamma A_\mu^*\bar\psi^\mu . \tag{A.8}$$

Now (A.2) and (A.3) coincide. The full construction is thus done in analogy to Section 3, iterating the check for the (co)chain map property and adjusting the definition of $\iota$ accordingly.

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
