# Peer review of "Yang-Mills Theory and the $\mathcal{N}=2$ Spinning Path Integral"

_SciPost Physics_

## Round 1 · Referee Report · Anonymous (Referee 1) · 2025-12-13

Report

This paper addresses the construction of a spacetime Yang–Mills action starting from an N=2 worldline model. The aim is to mimic, in a simplified setting, the more intricate case of string theory and the corresponding construction of string field theory based on an N=(1,1) worldsheet and its supermoduli space.
To this end, the authors propose a (non-unique) operator–state correspondence for the one-dimensional worldline and construct interaction vertices by defining integral forms on supermoduli space. These forms involve somewhat arbitrary choices of Poincaré duals, analogous to the RNS picture-changing operators in string theory.
The constructions presented in this paper could prove useful for analogous developments in superstring field theory. For this reason, I am inclined to recommend publication. However, before publication, I think that the paper would benefit from some improvements and clarifications, as outlined in the following paragraph.

Requested changes

1) The paper is highly technical and frequently refers to earlier, equally technical results. It would therefore be helpful to briefly summarize some background material and notation. For instance, the notation used for the vacuum state in Eq. (2.5) should be explained, and a short description or recap of the supercomplex of pseudo-forms employed throughout the paper would be useful. This additional material is needed to improve the communication of the paper’s results to the string theory community, and in particular to the string field theory (SFT) community, which appears to be one of the main target audiences of this work. Along the same lines, it would be helpful to explicitly define the Hamiltonian appearing in Eq. (2.4) ( is it simply p^2?).

2) The highly nonlocal operator d/□ appearing in Eq. (3.12) is somewhat troubling, and further discussion is needed to clarify how such an expression can be defined in a precise and well-controlled framework. Ideally, it would be preferable to avoid such singular expressions altogether.

3) The referee finds it surprising that an N=2 constraint algebra, as encoded in the BRST charge in Eq. (2.4), ultimately leads to essentially the same Yang–Mills theory obtained from an N=1 constraint algebra, as discussed for example by the authors themselves in Ref. [23]. This outcome appears to be related to the choice of picture-changing operator(s) Y in the N=2 setting. However, a clear discussion of how the results of Ref. [23] are connected to the present N=2 framework is either missing or only sketched. Such a discussion would be valuable, especially in view of the lessons for superstring theory that one expects to extract from these simplified models.

Recommendation

Ask for major revision

---

## Round 1 · Referee Report · Anonymous (Referee 2) · 2025-12-22

Disclosure of Generative AI use

The referee discloses that the following generative AI tools have been used in the preparation of this report:

polishing the sentences and grammar/typo check.
GPT-5.2.

Strengths

  1. The paper presents detailed computations in a relatively explicit which helps the reader verify the main claims.
  2. The perspective is conceptually original, in particular in how the Yang–Mills action is recovered via supermoduli integration and an appropriate choice of integration cycle.

Weaknesses

  1. The review/background material is relatively brief and can be difficult to follow without repeatedly consulting several of the cited references.

Report

This manuscript studies how to reproduce the standard Yang–Mills action starting from a spinning particle with $N=2$ worldline supersymmetry. The authors’ broader purpose is to implement a second-quantization procedure in close analogy with superstring field theory, where interaction terms in the string field action are obtained by integrating over appropriate regions in the moduli space of super Riemann surfaces. In the present $N=2$ spinning particle setting, the relevant geometric structure is the moduli space of a punctured superline.
From an algebraic viewpoint, the paper analyzes deformations of the \(A_\infty\) algebra associated with the free spinning particle and shows how these deformations match the Yang–Mills action. After an introduction and review, Section 3 constructs a suitable worldline state–operator map, which is not automatic unlike on the string-theory worldsheet. Section 4 and subsequent sections formulate the spacetime action as an integral over supermoduli; a key point emphasized is that one must choose a specific integration cycle in supermoduli space in order to recover the correct Yang–Mills action.
Overall, the manuscript provides new insights (substantially expanding on [16]) into the relationship between Yang–Mills theory and perturbative QFT from a worldline/supermoduli perspective. I expect it to be of interest to researchers working on related topics, and I recommend publication in SciPost Physics after the authors address the (optional) suggestions below.

Requested changes

  1. Throughout the manuscript the color index seems to be mostly suppressed. However, I could not consistently follow at which step(s) it should be introduced. It would improve readability if the authors clarify their convention once (e.g. early in Section 2) and indicate explicitly where color indices are being suppressed.
  2. In Eq. (2.5), the notations $1_{\psi}$, $1_c$, and $1_{\gamma}$ do not seem to be explained. Please define these objects when they first appear.
  3. On page 3, the manuscript states: “Therefore, the non-linear Yang–Mills equation cannot be written as a MC-equation on $V^{(0,-1)} 1_{\psi}$.” As written, this is confusing: $V^{(0,-1)} 1_{\psi}$ does not appear to be equipped with an algebra structure (rather, it looks like a module), so the phrase “MC-equation on $V^{(0,-1)} 1_{\psi}$” is not clearly meaningful a priori. Please clarify what structure is intended here (e.g. an $L_\infty$ or $A_\infty$ structure on some graded space containing this component) and in what precise sense the Yang–Mills equation fails to be expressible as a Maurer–Cartan equation in that setting.
  4. On page 4, in the line immediately below (3.1), I believe $\iota(V^{(0,1)})$ should read $\iota(V^{(0,1)} 1_{\psi})$. Please check and correct if appropriate.
  5. The paper by Ohmori and Tachikawa, arXiv:1303.7299, discusses how a Poincaré dual of ordinary moduli space sits inside supermoduli space arised in the case of Berkovits–Vafa embedding $N=0 \to N=1$. While the setup differs from the present $N=2$ worldline case, it might be worth citing and/or briefly commenting on possible conceptual connections.

Recommendation

Publish (easily meets expectations and criteria for this Journal; among top 50%)

---

## Editorial Decision

awaiting_resubmission